# Antiarrhythmic Properties of *Elsholtzia ciliata* Essential Oil on Electrical Activity of the Isolated Rabbit Heart and Preferential Inhibition of Sodium Conductance

**DOI:** 10.3390/biom10060948

**Published:** 2020-06-23

**Authors:** Regina Mačianskienė, Lauryna Pudžiuvelytė, Jurga Bernatonienė, Mantė Almanaitytė, Antanas Navalinskas, Rimantas Treinys, Inga Andriulė, Jonas Jurevičius

**Affiliations:** 1Institute of Cardiology, Medical Academy, Lithuanian University of Health Sciences, Sukilėlių pr. 15, LT-50161 Kaunas, Lithuania; regina.macianskiene@lsmuni.lt (R.M.); mante.almanaityte@lsmuni.lt (M.A.); antanas.navalinskas@lsmuni.lt (A.N.); rimantas.treinys@lsmuni.lt (R.T.); inga.andriule@lsmuni.lt (I.A.); 2Institute of Pharmaceutical Technologies, Medical Academy, Lithuanian University of Health Sciences, Sukilėlių pr. 13, LT-50161 Kaunas, Lithuania; lauryna.pudziuvelyte@lsmuni.lt (L.P.); jurga.bernatoniene@lsmuni.lt (J.B.); 3Department of Drug Technology and Social Pharmacy, Medical Academy, Lithuanian University of Health Sciences, Sukilėlių pr. 13, LT-50161 Kaunas, Lithuania

**Keywords:** *E. ciliata* essential oil, natural antiarrhythmic medicine, Langendorff-perfused heart, pseudo ECG, action potential, optical mapping

## Abstract

*Elsholtzia ciliata* essential oil (*E. ciliata*) has been developed in Lithuania and internationally patented as exerting antiarrhythmic properties. Here we demonstrate the pharmacological effects of this herbal preparation on cardiac electrical activity. We used cardiac surface ECG and a combination of microelectrode and optical mapping techniques to track the action potentials (APs) in the Langendorff-perfused rabbit heart model during atrial/endo-/epi-cardial pacing. Activation time, conduction velocity and AP duration (APD) maps were constructed. *E. ciliata* increased the QRS duration and shortened QT interval of ECG at concentrations of 0.01–0.1 μL/mL, whereas 0.3 μL/mL (0.03%) concentration resulted in marked strengthening of changes. In addition, the *E. ciliata* in a concentration dependent manner reduced the AP upstroke dV/dt_max_ and AP amplitude as well as APD. A marked attenuation of the AP dV/dt_max_ and a slowing spread of electrical signals suggest the impaired functioning of Na^+^-channels, and the effect was use-dependent. Importantly, all these changes were at least partially reversible. Our results indicate that *E. ciliata* modulates cardiac electrical activity preferentially inhibiting Na^+^ conductance, which may contribute to its effects as a natural antiarrhythmic medicine.

## 1. Introduction

Antiarrhythmic drugs are of particular importance, but many of them are discarded due to side effects [1,2]. Therefore, the discovery of new ones is of particular importance. Voltage-gated Na^+^-channels play an important role in the initiation and conduction of cardiac electrical signals; therefore, selective inhibition of Na^+^ conductance could be a promising antiarrhythmic target [3,4].

It is believed that natural herbal medicines may have milder side effects. One of such medicines can be *Elsholtzia ciliata* (Lamiaceae) essential oil (further *E. ciliata*). This herbal preparation has been developed in Lithuania and is supposed to be harmless.

Initially, a chemical composition of *E. ciliata* prepared by different extraction methods from fresh/frozen/dried herbal materials has been described: a total of 48 compounds were identified by gas chromatography-mass spectrometry method [5]. The major compounds of this essential oil were dehydroelsholzia ketone (78.28%) and elsholtzia ketone (14.58%) (Appendix A). The composition of this essential oil was similar to that reported by other groups [6,7,8], with the main component dehydroelsholtsia ketone, which was first isolated from this source in 1997, and for which a structure of 3-methyl-2-(3-methylbut-2-enoyl)furan was suggested [9]. Chemical composition of *E. ciliata* herb could vary because of environmental factors and different growing places [10]. Furthermore, an anticancer activity of the *E. ciliata* on human glioblastoma (U87), pancreatic cancer (Panc-1) and triple negative breast cancer (MDA-MB231) cell lines has been demonstrated in vitro [5].

Recently, the *E. ciliata* has been internationally patented by Bernatoniene et al. [11] as having antiarrhythmic properties characteristic to the first-class antiarrhythmic drugs. However, a detailed mechanism of *E. ciliata* action in the heart with tissue architecture and properties remaining intact and thus allowing a comprehensive analysis of cardiac electrophysiology in (nearly an) in vivo setting has not been described. The clinically relevant concentrations of the compound remain unclear in terms of safety of use and efficacy of cardioprotection, because many medicines, including essential oil accumulating herbal materials, have side effects at high concentrations.

In the present study, using the Langendorff-perfused rabbit heart model, we studied effects of *E. ciliata* on the cardiac surface ECG and action potentials (APs) during atrial, endo- and epi-cardial stimulation. In particular, we tested how the effects of *E. ciliata* relate with the first-class antiarrhythmic drug features. Our study is the first to examine the role of *E. ciliata* in modulation of cardiac electrical activity and to link these cellular mechanisms to the properties typical for the first-class antiarrhythmic drugs.

## 2. Materials and Methods

### 2.1. Plant Material

The essential oil from *E. ciliata* dried herb (purchased from “Žolynų namai“, Vilnius, Lithuania; a voucher specimen #L 17710) was prepared using a Clevenger distillation apparatus. A sample of dried grounded herb (30 g) was mixed with 500 mL purified water and submitted to extraction for 4 h at 120 °C [5]. The composition analysis of *E. ciliata* essential oil obtained from dried herb by hydrodistillation method showed 26 components. [5]. In this study, according to gas chromatography–mass spectrometry results, the main sesquiterpenes obtained were: beta-bourbonene (0.57%), isocaryophyllene (0.57%), beta-cubebene (0.06%), ledene (0.05%), alpha-caryophyllene (1.84%), alpha-cubebene (0.02%), germacrene D (0.24%), trans-alpha-bergamotene (0.55%), alpha-farnesene (0.66%), gamma-cadinene (0.15%), and delta-cadinene (0.28%) [5]. Dehydroelsholtzia ketone (78.28%) and elsholtzia ketone (14.58%) were the main compounds in this essential oil [5]. According to this data, sesquiterpenes made 4.99% and ketones 92.86% of the total composition of the essential oil [5].

### 2.2. Chemicals

All other reagents were from Sigma-Aldrich (Schnelldorf, Germany), excluding near infrared (NIR) voltage-sensitive dye (VSD), di-4-ANBDQBS (JPW6033) (AAT Bioquest, Sunnyvale, CA, USA) and (±) blebbistatin (Cayman, Ann Arbor, Michigan, USA). Water insoluble compounds were initially dissolved in dimethyl sulfoxide (DMSO) or ethanol to make stock solutions, which were then diluted. A stock solution of *E. ciliata* (1:1 in ethyl alcohol, anhydrous, ≥99.8%) was freshly prepared prior to each experiment and diluted to various concentrations (0.01–0.3 µL/mL) with Tyrode solution just before use. The highest concentration of the solvent was <0.1%, and the solvent did not affect the measurements [12].

### 2.3. Heart Preparation

This study was carried out in accordance with the European Community guiding principles. Experiments on New Zealand white rabbits were approved by the State Food and Veterinary Service of the Republik of Lithuania (No. G2-34, 24 September 2015).

New Zealand white rabbits (n = 11) of either sex (~3.5 kg) were used in the study. The excised hearts were cannulated via its aorta and swiftly connected to a Langendorff system [12]. Hearts were perfused retrogradely at constant pressure (80 mmHg) for 30 min with oxygenated (100% O*_2_*) Tyrode solution (in mM/L: 135 NaCl, 5.4 KCl, 1.8 CaCl*_2_*, 0.9 MgCl*_2_*, 0.33 NaH*_2_*PO_4_, 10 glucose, and 10 HEPES; pH 7.4 at 37 ± 0.5 °C). Then the perfusion was switched to a recirculation mode and blebbistatin (10 μM/L) was added to inhibit contraction, in order to eliminate heart movement during recordings of the action potentials (APs). After stabilization, the hearts were stained with NIR VSD, di-4-ANBDQBS, which was added into the perfusate at a final concentration of 3 µM/L. Then *E. ciliata* was added to the perfusate to obtain the final concentrations of 0.01, 0.03, 0.1, and 0.3 (µL/mL) (recalculated as percent concentrations: 0.001%, 0.003%, 0.01%, and 0.03%, respectively). Recordings were obtained after 10 min exposure to each cumulatively added *E. ciliata* concentration.

Atrial and epicardial pacing was performed via bipolar hook electrodes embedded in the atrium and epicardial surface of left ventricle (near the base), respectively. Endocardial pacing was performed via a bipolar silver electrode inserted into the left ventricular cavity close to the apex. The heart was continuously electrically stimulated for a 300 ms period, with a 2 ms pulse width set at twice the threshold. When pacing was initiated via atrial/endo-/epi-cardial surfaces, the spread of electrical activity varied, creating different situations in the myocardium [13].

### 2.4. Registration of Electrical Activity

A pseudo electrocardiogram (pseudo ECG) was recorded via two electrodes placed on the posterior surface of the heart at a distance of about 10 mm, using LabChart8 Pro software (ADInstruments, Oxford, UK). Simultaneously, the electrical activities registered with microelectrodes and with NIR VSD were obtained from the anterior surface of the heart.

Microelectrode-APs were recorded using glass microelectrodes (filled with 3 M KCl), which were inserted in the left and right ventricular (LV and RV) walls from the epicardial surface. The recorded APs were amplified and digitized by the 16 channel PowerLab system (ADInstruments, Oxford, UK) at a frequency of 20 kHz.

Use-dependent effect of *E. ciliata* was detected from the decrease in AP upstroke velocity (dV/dt_max_) [14], when stimulation rate was switched from 2.5 Hz to 5 Hz. The kinetics of attenuation of dV/dt_max_ was determined by fitting the dV/dt_max_ decrease in time with an exponential decay function. The time constant (*τ*) of block development was expressed by time necessary to achieve 63.2% of the maximal steady-state decrease of dV/dt_max_.

Optical-APs (OAPs) were obtained using NIR VSD, di-4-ANBDQBS. The dye was excited with collimated light using a 660 nm LED (M660L3, filtered at 650/40; from Thorlabs, Newton, New Jersey, USA). The emitted fluorescence was filtered by a 720 nm long pass filter (NT46-066, Edmund Optics, Barrington, New Jersey, USA), which was placed in front of the camera. Optical movies were obtained with a fast (500 frames per second, and 128 *×* 128 pixels), 14 bit EMCCD camera (iXon^EM^+DU 860, Andor Technology Ltd, Belfast, Northern Ireland, UK). The anterior surface of the heart was imaged (captured LV, RV and the interventricular groove), the field of view was 20 *×* 20 mm. The OAPs were taken from an area of 5 *×* 5 pixels. The background fluorescence (F) was subtracted from every frame of the recording. The optical signal was normalized with respect to the background fluorescence to obtain the voltage dependent fractional change in the fluorescence signal (ΔF/F).

The pseudo ECG and AP were recorded and analyzed using LabChart8 Pro software. Optical movies were processed using ImageJ (developed by Wayne Rasband at the NIH and the LOCI, University of Wisconsin, Madison, Wisconsin, USA) software.

### 2.5. Data Analysis and Statistics

Time-to-R_max_ was detected as time to maximal amplitude to the R-peak from the stimulus. Changes in QRS and QT intervals were automatically recorded and calculated using LabChart8 Pro software ECG analysis for rabbit heart. QRS and QT intervals were taken as the time interval from Q wave to the S wave or 0.5 of T wave decay, respectively.

The activation time (AT) was detected as the time interval from stimulus to dV/dt_max_ of AP upstroke or 50% depolarization of the OAP. OAP upstroke duration was calculated between 20 and 80% of depolarization (UD20-80). Conduction velocity and conduction vector maps were constructed from AT data using Scroll 1.16 software. The duration of the AP and OAP (APD and OAPD) was detected at the level of 20%, 50%, and 90% repolarization, calculated from their AT. The OAP maps were constructed using custom Scroll 1.16 software developed by Dr. S. Mironov (University of Michigan, Ann Arbor, Michigan, USA).

Data are presented as the mean ± standard error of the mean (S.E.M.). The significance of differences was evaluated using one-way analysis of variance (ANOVA). A value of *p* < 0.05 was considered statistically significant.

## 3. Results

### 3.1. Effects of E. ciliata on Pseudo ECG

To demonstrate how *E. ciliata* changes electrical activity, we applied it on the isolated perfused rabbit heart. The effect of that herbal preparation was investigated by recording pseudo ECG from the posterior surface of epicardium. Typical examples of pseudo ECGs, as presented in Figure 1A, were obtained after perfusion with the Tyrode solution (control) followed by a 10 min perfusion with each of 0.01, 0.03, 0.1 and 0.3 µL/mL concentrations of the *E. ciliata*, and a washout period (~20 min), when stimulated from the epicardial surface with a period of 300 ms.

Figure 1B shows real-time recordings, from the same experiment as in Figure 1A, during progressive change in pseudo ECG signals, with the modification by four *E. ciliata* concentrations (as indicated). Pseudo ECG records were converted to “pseudo-colors” and overlaid onto the color 3D image of views from both top and bottom, as presented in Figure 1B (upper and lower panels, respectively). Notice the transient changes on surface pseudo ECG recordings, from front (white arrow) to back, associated with the QRS widening and QT interval shortening resulting from the addition of *E. ciliata*, and the recovery upon washout.

Our results revealed that *E. ciliata* in a concentration-dependent manner lengthens both the time-to-R_max_ and the QRS duration, while shortening the QT interval. The low concentrations of *E. ciliata* (≤0.1 µL/mL) only moderately affected the time-to-R_max_, QRS, and QT interval, but the effects were statistically significant (n = 5, *p* < 0.05). Of note, an abrupt increase, as presented in Figure 1C, was detected when applying *E. ciliata* concentrations equal or higher (not illustrated) than 0.3 µL/mL. In addition, atrioventricular (A-V) blocks were observed when using 0.3 µL/mL concentration. Action of the *E. ciliata* on the pseudo ECG was at least partially reversible upon washout with normal Tyrode solution. The mean values of pseudo ECG parameters obtained with increasing *E. ciliata* concentrations are presented in Table 1.

### 3.2. Effects of E. ciliata on Action Potential

It is commonly known that QRS express propagation features of the electrical signal in the heart, and QT interval reflects the AP duration [15,16]. The QRS and QT intervals can be used to evaluate the conduction velocity and APD, respectively. The data presented in Figure 2A,B show that APs, obtained with standard glass microelectrodes inserted in the LV wall from the epicardial surface of the rabbit heart, follows the changes in the pseudo ECGs depending on the modifications due to the *E. ciliata*. APs were recorded during stimulation from both endo- and epi-cardial surfaces of ventricles, and when pacing was applied from the atrium. During atrial stimulation, as described previously by Macianskiene et al. [13], the spread of electrical excitation reaches the Purkinje fibers first, therefore activation of the entire endocardium of the ventricles occurs almost simultaneously. Further, the electrical wave propagates transmurally from the endocardium-to-epicardium. Upon endocardial stimulation, due to location of pacing electrode close to the apex, the excitation travelling upward from the apex to the base of ventricles was observed. While stimulating from the epicardium, when the location of the pacing-electrode on the left side of the LV was close to the base, the propagation of the excitation wave was almost parallel to the epicardial surface and of an elliptical shape, due to the fiber orientation in the LV. In this study, effects of *E. ciliata* were investigated using all three pacing types.

Figure 2C displays values of the maximal velocity of APs upstroke (dV/dt_max_) showing a dose-dependent inhibition with *E. ciliata*, when the heart was stimulated from epicardium. All concentrations of *E. ciliata* decreased the amplitude of dV/dt_max_, suggesting inhibition of Na^+^-channels, and caused an increase in the activation time (AT).

As shown in Figure 3 and Appendix A, under control conditions, the AT was longest during stimulation from the atrium, possibly due to the slow excitation propagation via the atrioventricular (A-V) node where AP are generated by L-type Ca^2+^-current. While upon the action of *E. ciliata*, the activation time (AT) increased at all pacing types. However, when the heart was stimulated from the atrium, the effect on the AT using a 0.3 µL/mL concentration of the *E. ciliata* was much bigger, possibly, because of inhibition of the L-type Ca^2+^-current. As a consequence, the electrical excitation propagation via the A-V node slowed down. Besides, in all experiments the A-V blocks were caused after >5 min, when 0.3 µL/mL concentration of the *E. ciliata* was applied.

Figure 3C displays mean values of the change in AP duration at 20%, 50% and 90% repolarization (APD20, APD50, and APD90, respectively) (see also Appendix A) obtained upon action of the *E. ciliata* during atrial/endo-/epi-cardial stimulation. After application of each concentration of the *E. ciliata,* a decrease in APD at all repolarization levels was observed, but the effects were at least partially reversible upon washout.

### 3.3. Use-Dependent Effect of E. ciliata on the dV/dt_max_ of the AP

It is typical, that the mode of action and efficacy of the antiarrhythmic preparations or local anesthetics, that slow down depolarization of the AP upstroke and inhibit Na^+^-channels, are highly dependent on their specific interaction with Na^+^-channels, depending on the channel state: open or closed, activated or inactivated. Therefore, the effectiveness of these preparations is use-dependent, and their activity in the heart may depend on stimulation rate, resting potential, and AP duration. Using the procedure to examine the use-dependent effects, i.e., changing the pacing cycle length from 400 to 200 ms, as presented in Figure 4, we revealed that *E. ciliata* (0.1 μL/mL) extra reduced the maximal velocity of AP upstroke at an increased pacing rate (Figure 4D).

In the control, an increase in the pacing rate slightly reduced both the AP amplitude and duration, and slowed the dV/dt_max_ down by 10.7 ± 2.1% with time constant (**τ**) of 2.37 ± 0.09 s (n = 7) (Figure 4A,C). The *E. ciliata* of 0.1 µL/mL concentration, as presented in Figure 4B,D, markedly reduced dV/dt_max_ if compared with the control. The kinetics of the AP upstroke dV/dt_max_ depression was evaluated with an exponential approximation of the dV/dt_max_ changes as a function of time. The time constant (**τ**) calculated for the block development in each experiment, as presented in Figure 4D, insert, was 0.55 s (on average was 0.63 ± 0.11), and dV/dt_max_ was significantly reduced by 32.9 ± 4.8% s (n = 7, *p* < 0.05). These data indicate that the *E. ciliata* slowed the maximal velocity of AP upstroke depolarization down much stronger at a higher pacing rate, and the reaction of dV/dt_max_ change is fast (**τ** < 1 s).

### 3.4. Effects of the E. ciliata on Optical Action Potentials

The data above show that the *E. ciliata* caused concentration-dependent modification of both pseudo ECG (Figure 1) and AP (Figure 2). To extend our observations, we evaluated effects of the *E. ciliata* on the optically recorded APs (OAPs) using a NIR VSD, di-4-ANDBQBS, on the rabbit hearts. Figure 5A–C shows typical examples of overlapped OAP recordings obtained during atrial/endo-/epi-cardial stimulation. Data show that the activation time of the OAPs and their shape, similarly as with the microelectrode-recorded electrical APs (Figure 2), change upon action of the *E. ciliata* of the increasing concentrations.

Figure 6 shows summary data of the concentration-dependent changes of OAP activation time, the voltage-sensitive fraction of fluorescence, OAP upstroke duration, and OAP duration after application of the *E. ciliata* (0.01–0.3 μL/mL) under different pacing types (atrial/endo-/epi-cardial). Also, the mean values of OAP parameters, obtained under three pacing types, are presented in Appendix A. These data demonstrate that independently of the registration technique, using the standard microelectrodes, as presented in Figure 2 and Figure 3, or the optical mapping, as shown in Figure 5 and Figure 6, the *E. ciliata* increased the AT (Figure 3A and Figure 6A), slowed the depolarization of the AP upstroke down (Figure 3B and Figure 6C), and shortened the time of the electrical excitation (Figure 3C and Figure 6D). Besides, the data show that *E. ciliata* in a concentration-dependent manner reduced the voltage-sensitive fraction of fluorescence (Figure 6B). That effect, at least partly, could be explained by the lowered amplitude of the AP (see Appendix A) because of a possible blocking action of the *E. ciliata* on the Na^+^ and Ca^2+^ currents. Reduction of ΔF/F also might happen because of summation of OAPs from many cells located at different depth and spatially, which amplitudes are separated in time.

### 3.5. Effects of the E. ciliata on the Spatial and Temporal Dynamics of the OAP

The representative maps of electrical activity recorded on the whole heart during epicardial stimulation (300 ms period) using the optical mapping demonstrate how the activation time, conduction velocity and the direction of conduction change, as presented in Figure 7, as well as the OAP duration change at 20%, 50%, and 90% of repolarization (Figure 8), obtained under the control conditions and after treatment with the *E. ciliata* of 0.01, 0.03, 0.1, and 0.3 μL/mL concentrations. Constructed conduction velocity maps demonstrate proportional changes under both control conditions and after application of *E. ciliata*, and that there was no increase in the dispersion of this parameter. Mean values of conduction velocity in control and after *E. ciliata* of a 0.1 µL/mL concentration are of 0.77 ± 0.06 and 0.50 ± 0.03, respectively, at epicardial stimulation (n = 5, *p* < 0.05), and of 1.04 ± 0.08 and 0.49 ± 0.05, respectively, at endocardial stimulation (n = 6, *p* < 0.05) (for concentration-dependence data see in Appendix A).

Under the control conditions and at low concentrations of *E. ciliata* extracts, as presented in Figure 7C, the direction and length of conduction vectors vary, especially at the bottom of conduction vector maps. However, at high concentrations, all vectors became of the same length and oriented in the same direction and this indicated harmonization in impulse propagation. Such uniform direction and the spread of electrical impulse propagation at high concentrations indicate that *E. ciliata*, alongside with the antiarrhythmic features, might have a less pro-arrhythmic effect.

However, as was mentioned above, the direction of the electrical excitation propagation depends on the stimulation type. Therefore, the spatial and temporal dynamics of the OAP under the *E. ciliata* action, when stimulating from the atrium (Appendix A) or endocardium (Appendix A), in corresponding maps are different.

## 4. Discussion

Antiarrhythmic drug therapy is the primary treatment for heart arrhythmias, but due to the potential proarrhythmic effects of the drugs, there has been interest in development of new antiarrhythmic drugs, including herbal drugs, which may have a milder and less proarrhythmic effect [17].

The majority of the antiarrhythmic drugs are blockers of various ion channels of the cardiac cells. Over 50 have years passed since Vaughan Williams created a classification of the antiarrhythmic drugs, which is still valuable and, of course, elaborated [18]. Certainly, in the classification blockers of different ion channels, which are also modulators of different phases of AP, are assigned to different classes of antiarrhythmic preparations. The most widely used and the biggest is the Class 1, i.e., the Na^+^-channels blockers, which slow the depolarization of the AP upstroke and reduce velocity of the electrical impulse propagation in the myocardium. Additionally, the Class 1, according to their action on the AP duration, was subdivided by Vaughan Williams into the three subclasses: compounds from the subclass 1A prolong the AP, 1B shorten the AP, and 1C negligibly change the AP duration. Further investigations demonstrated that the antiarrhythmic compounds of Class 1 differ according to their use-dependent effect, i.e., their inhibition of the Na^+^-current or dV/dt_max_ is dependent on the time and depolarization of the cells or on the pacing rate. According to the modulated receptor hypothesis [19] this different dependence of the activity of antiarrhythmic drugs on time and membrane potential might be explained by the different interaction of those drugs with Na^+^-channels in their different states of activity: open or closed, and activated or inactivated. Class 1A antiarrhythmic compounds have been shown to interact better and block Na^+^-channels in the activated or open state which shortly last only during the AP upstroke, so their efficacy is not very dependent on the AP duration, and their interaction kinetic time constant (**τ**) ranges from 1 to 10 s. Class 1B antiarrhythmic drugs highly interact and block Na^+^-channels in the long lasting inactivated state during entire AP, so their efficacy is remarkably dependent on the duration of AP, and the constant (**τ**) of the kinetics of their interaction is less than 1 s. Meanwhile, class 1C antiarrhythmic drugs interact well and block Na^+^-channels in both the activated and inactivated states, so their effectiveness also depends on the APD, but their interaction is very slow, the kinetics constant (**τ**) is more than 10 s [14,20].

Our studies showed that the *E. ciliata* exerts inhibition on the dV/dt_max_, and blocking kinetics obtained by increased stimulation frequency of the heart had a time constant that was less than 1 s. Based on these results, it can be stated that the *E. ciliata* blocks the Na^+^-channels, inducing fast inhibition, preferably interacting with the channels in their inactivated state. In addition, the *E. ciliata* shortened the AP. All these data possibly enable us to state that the *E. ciliata* possesses features that are typical of class 1B antiarrhythmic compounds.

The *E. ciliata* also has other properties characteristic of Class 1 antiarrhythmic drugs, it prolongs myocardial activation time, slows the velocity down of electric impulse propagation, widens the QRS interval and shortens the QT interval. In addition, it can be noted that applying higher *E. ciliata* concentrations, harmonization of the impulse propagation directions (vectors) is observed (Figure 7), which could be one more of the assumptions of *E. ciliata* antiarrhythmic efficiency.

In most of our experiments, we used four concentrations of *E. ciliata*, from 0.01 to 0.3 µL/mL; all of them had a concentration dependent effect on the cardiac electrophysiological properties. The effect was observed applying all concentrations, apparently due to the *E. ciliata* blockade of Na^+^-current. Using the concentration of 0.3 µL/mL and when the heart was paced from the atria, impulse propagation blocks were recorded. It should be noted that when the heart is stimulated from the atria, in the impulse propagation pathway there is an A-V node, generating APs, the upstroke of which is formed, and at the same time the propagation is regulated by L-type Ca^2+^-current. It is possible that the *E. ciliata* at high concentrations inhibits not only the Na^+^-current, but also the Ca^2+^-current, and in such way slows the impulse propagation down via A-V node inducing conduction block. We demonstrated that the *E. ciliata* shortened the APs, but it is known that the APs duration is almost independent of Na^+^-current. Our research does not allow us to determine which current changes dominates in the AP shortening. The AP may be shortened due to the modulation of Ca^2+^ and K^+^ currents by *E. ciliata* [21]. The current generated by Na/Ca-exchange system the strength of which depends on the activity of the intracellular Na^+^, could be changed due to the reduction of Na^+^-current [22] during the action of *E. ciliata* and, accordingly, also might participate in regulation of the AP duration.

Summarizing it can be said that the *E. ciliata* possesses features of class 1B antiarrhythmic preparations and, supposedly, might be effective in the treatment of the ventricular tachycardias.

## 5. Conclusions

This study provides the first characterization of the effects of the *E. ciliata* essential oil in a Langendorff-perfused heart model for the registration of cardiac electrical activity. *E. ciliata* increased QRS duration, shortened the QT interval of pseudo ECG, and reduced the action potential upstroke dV/dt_ma_, and the duration. A marked attenuation of the AP dV/dt_max_ and slowing down of the spread of electrical signals suggested an impaired functioning of the Na^+^-channels, and the effect was use-dependent. Our results demonstrate that *E. ciliata* possesses features of class 1B antiarrhythmic preparations and, supposedly, might be effective in the treatment of the ventricular tachycardias.

## 6. Patents

Bernatonienė, J.; Pudžiuvelytė, L.; Jurevičius, J.; Mačianskienė, R.; Šimonytė, S. *Elsholtzia ciliata* essential oil extract as antiarrhythmic drug. WO Patent; Application date: 6 April 2018 = Extrait d’huile essentielle d’Elsholtzia ciliata comme médicament antiarythmique/Lithuanian University of Health Sciences. WO/2019/193400, 10 October 2019.

## Figures and Tables

**Figure 1 biomolecules-10-00948-f001:**
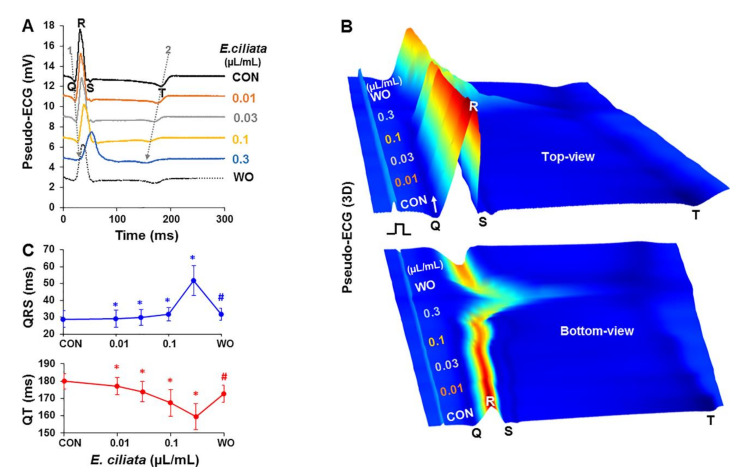
*E. ciliata* effects on the pseudo ECGs of Langendorff-perfused rabbit hearts. Epicardial pacing with a 300 ms period was applied. (**A**) Superimposition of steady-state pseudo ECG traces under control conditions (CON; black) and after application of 0.01, 0.03, 0.1 and 0.3 µL/mL concentrations of *E. ciliata* (orange, grey, yellow and blue, respectively) for 10 min each, followed by perfusion without *E. ciliata* (WO; dotted). Q, R, S, and T waves indicated. Notice a lengthened time-to-R_max_ and shortened QT interval as indicated by dotted arrows 1 and 2, respectively. (**B**) Three-dimensional (3D) view of pseudo ECGs, from front to back, showing time-dependent effect of increasing concentrations of *E. ciliata* from the same data as in (A), constructed by LabChart8 Pro software ECG analysis plot function, in which the X, Y and Z axis represent location, time, and height of the detected peaks of the pseudo ECG, respectively. Wave amplitudes are indicated by colors from blue to red. (**C**) Mean data of QRS (upper) and QT (lower) interval change when applying different *E. ciliata* concentrations as indicated (* *p* < 0.05 for *E. ciliata* vs. control; ^#^
*p* < 0.05 for washout vs. *E. ciliata* of 0.3 µL/mL concentration; n = 5 for each).

**Figure 2 biomolecules-10-00948-f002:**
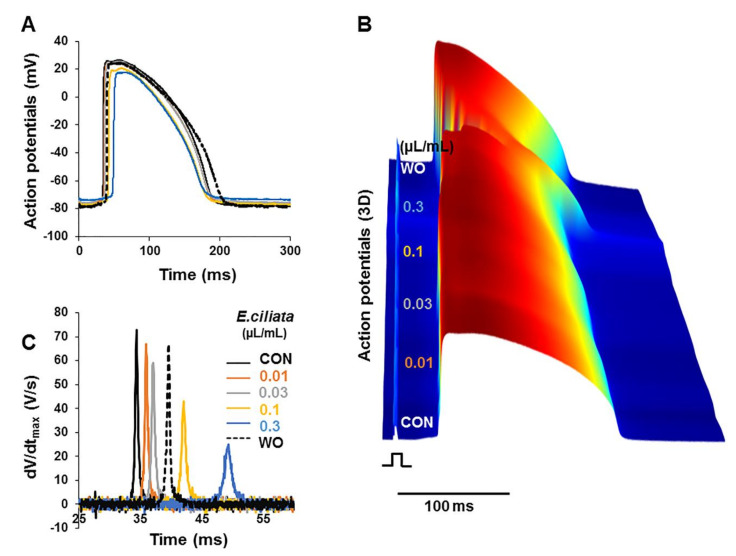
*E. ciliata* effects on the action potentials (APs) of Langendorff-perfused rabbit heart. Epicardial pacing with a 300 ms period was applied. (**A**) Superimposition of representative traces of APs from microelectrodes, from the same experiment as in Figure 1, under control conditions (CON; black) and after application of 0.01, 0.03, 0.1, and 0.3 µL/mL concentrations of *E. ciliata* (orange, grey, yellow and blue, respectively), followed by washout without the *E. ciliata* (WO; dotted). (**B**) Three-dimensional (3D) view of APs, from front to back, showing time-dependent effect of *E. ciliata* from the same data as in (A); 3D view construction details are the same as in Figure 1B. (**C**) Values of the maximal velocity of APs upstroke (dV/dt_max_) after application of *E. ciliata* concentrations as indicated from the same experiment as in (A).

**Figure 3 biomolecules-10-00948-f003:**
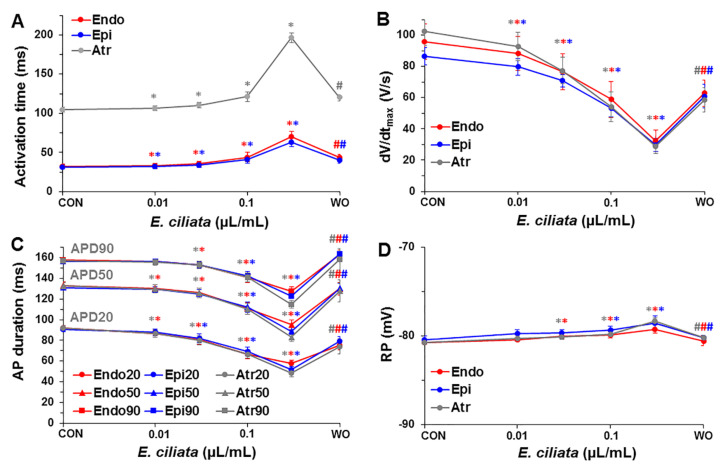
Dependence of the AP parameters on *E. ciliata* concentrations. Steady-state level measurements during atrial/endo-/epi-cardial stimulation (300 ms period) of (**A**) activation time, (**B**) the maximal velocity of AP upstroke depolarization (dV/dt_max_), (**C**) AP duration (APD) at 20%, 50% and 90% repolarization, and (**D**) resting potential (RP), measured under control conditions (CON) and at the end of 10 min perfusion with each of 0.01, 0.03, 0.1 µL/mL concentrations of the *E. ciliata* (except for 0.3 µL/mL, which was ~5 min), followed by a washout period (WO) (* *p* < 0.05 for *E. ciliata* vs. control; ^#^
*p* < 0.05 for washout vs. *E. ciliata* of 0.3 µL/mL concentration; n = 7 for each).

**Figure 4 biomolecules-10-00948-f004:**
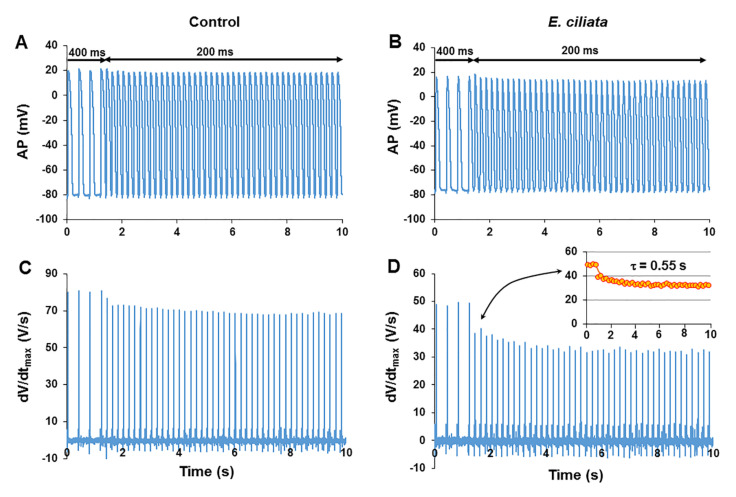
Monitoring the use-dependence effect of the *E. ciliata*. (**A**,**B**) APs and (**C**,**D**) dV/dt_max_ response to abrupt change of the pacing cycle length from 400 ms to 200 ms in the control and after action of the *E. ciliata* (0.1 μL/mL), respectively. Arrows on the horizontal bar indicate the moment of the pacing cycle length switch. Notice a stronger reduction of dV/dt_max_ at an increased stimulation rate after application of the *E. ciliata* vs. control. For detection of the kinetics of dV/dt_max_ depression see Methods, (D, *Insert*) experimental points of dV/dt_max_ fitted with exponential decay (line) with τ = 0.55 s.

**Figure 5 biomolecules-10-00948-f005:**
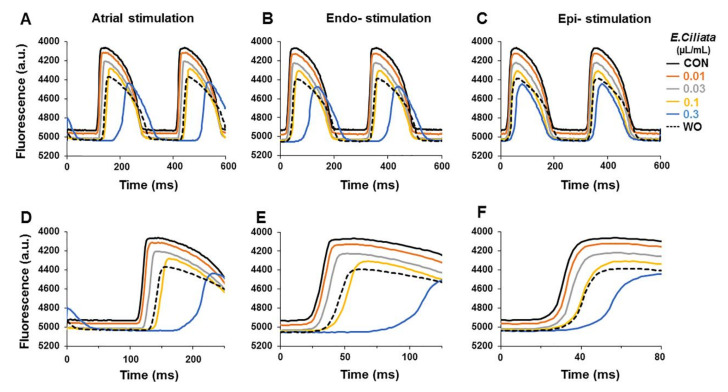
*E. ciliata* effects on optically recorded APs using voltage-sensitive fluorescent dye. Atrial/endo-/epi-cardial pacing with a 300 ms period was applied. (**A**–**C**) Superimposition of the optical APs (OAPs) and (**D**–**F**) their upstrokes on an expanded scale obtained in the control (CON; black) and after application of 0.01, 0.03, 0.1, and 0.3 μL/mL concentrations the *E. ciliata* (orange, grey, yellow and blue, respectively), followed by a washout period (WO; dotted). Notice a delay in activation time with 0.3 μL/mL concentration of the *E. ciliata*, and reversibility. Zero time indicates the time of stimulus.

**Figure 6 biomolecules-10-00948-f006:**
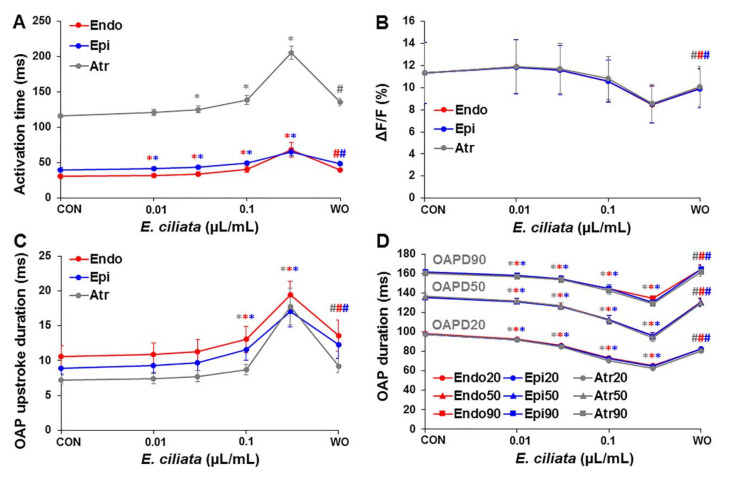
Dependence of the OAP on the *E. ciliata* concentrations. Steady-state level measurements during atrial/endo-/epi-cardial stimulation (300 ms period) of (**A**) activation time, (**B**) voltage-sensitive fraction of fluorescence (ΔF/F), (**C**) OAP upstroke duration (UD), and (**D**) OAP duration (OAPD) at 20%, 50% and 90% of repolarization, done under the control (CON) conditions and at the end of each 10 min perfusion with 0.01, 0.03, and 0.1 µL/mL concentrations of the *E. ciliata* (except for 0.3 µL/mL the perfusion was ~5 min), followed by a washout (WO) period (* *p* < 0.05 for *E. ciliata* vs. control; ^#^
*p* < 0.05 for washout vs. *E. ciliata* of 0.3 µL/mL concentration; n = 6 for each).

**Figure 7 biomolecules-10-00948-f007:**
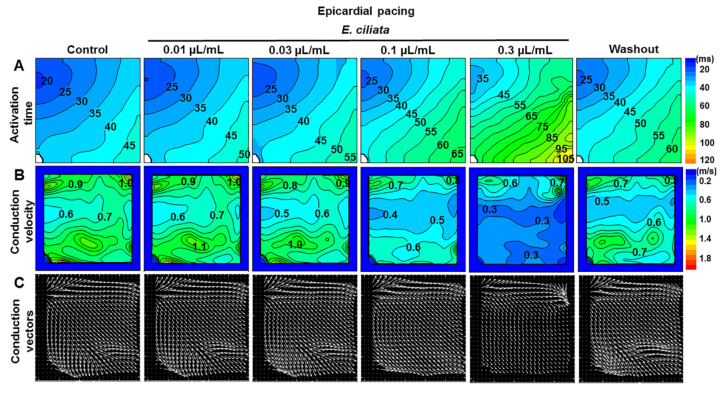
Optical maps revealing *E. ciliata* effects on electrical activation process in the heart during epicardial stimulation. A voltage-sensitive fluorescent dye, di-4-ANBDQBS, was used. The pacing period was 300 ms. (**A**) Activation time (in ms), (**B**) conduction velocity (in m/s), and (**C**) conduction vector maps in the control and after applying 0.01, 0.03, 0.1, and 0.3 µL/mL concentrations of the *E. ciliata*, followed by a washout period. The interval between isochrones is 5 ms for the activation time, 0.1 m/s for the conduction velocity. Note that the conduction vector length is proportional to the conduction velocity. The direction of the activation wave movement is from blue-green to red. The stimulation electrode was located on the left close to the LV base.

**Figure 8 biomolecules-10-00948-f008:**
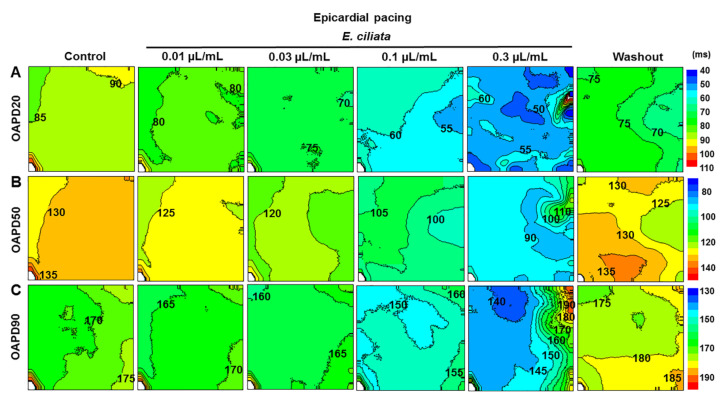
Changes in OAP duration maps induced by *E. ciliata* during epicardial stimulation. (**A**) OAPD20, (**B**) OAPD50, and (**C**) OAPD90 maps in the control and after applying 0.01, 0.03, 0.1, and 0.3 µL/mL concentrations of the *E. ciliata*, followed by a washout period. OAPD20, OAPD50, and OAPD90 maps calculated at 20%, 50%, and 90% of repolarization, respectively. The interval between isochrones is 5 ms. Other notations are the same as in Figure 7.

**Table 1 biomolecules-10-00948-t001:** The efficacy of *E. ciliata* essential oil action on pseudo ECG changes in the rabbit heart.

Concentration (µL/mL)	Time-to-R_max_ (ms)	QRS Interval (ms)	QT Interval (ms)
Control	37.7 ± 2.7	23.8 ± 2.5	179.9 ± 4.3
0.01	38.2 ± 2.6 *	24.1 ± 2.5 *	177.3 ± 5.0 *
0.03	39.0 ± 2.5 *	24.8 ± 2.5 *	174.0 ± 6.0 *
0.1	41.9 ± 2.5 *	27.0 ± 2.3 *	167.5 ± 7.7 *
0.3	59.2 ± 2.4 *	44.8 ± 4.7 *	159.6 ± 7.5 *
Washout	41.6 ± 2.4 ^#^	26.9 ± 2.1 ^#^	172.7 ± 5.0 ^#^

Epicardial pacing was set at 300 ms; Tyrode solution without *E. ciliata*, control and washout; Note: values are the mean ± S.E.M., n = 5 (for each); * *p* < 0.05 for *E. ciliata* vs. control; ^#^
*p* < 0.05 for washout vs. *E. ciliata* of 0.3 µL/mL concentration.

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
