# Peer review of "Antiarrhythmic Properties of *Elsholtzia ciliata* Essential Oil on Electrical Activity of the Isolated Rabbit Heart and Preferential Inhibition of Sodium Conductance"

_biomolecules, 2020, doi:10.3390/biom10060948_

Round 1
Reviewer 1 Report
This is a well done study of the electrophysiological effect of Elsholtzia ciliata essential oil on the rabbit heart showing convincingly that the plant extract reduces the action potential (AP) amplitude, AP upstroke and AP propagation velocity leading to the conclusion that the extract exerts blocking effects on voltage gated Na channels like Ib type antiarrhythmic drugs. At higher concentrations, the extract slows down the AP propagation from the atria to the ventricles, which is probably due to antagonistic effects on voltage gated Ca2+ channels also.
However, I do have some problems with the interpretation of the data by the authors.
- The authors state that the herbs extracts may be harmless regarding side effects of antiarrhythmic drugs. It is well known that INa blockade with slowing of the AP propagation is potentially pro-arrhythmogenic. In this regard, I do not see any difference in the pro-arrhythmogenic effects of the herbs extract and antiarrhythmic chemicals lice lidocaine. The authors must discuss it.
- Any information about the plasma concentrations of the orally administered drug is needed to range the concentrations used in the study with regard to the therapeutic and side effects.
- Contrary to the statement of the authors, the measured electrophysiological effects are not, at least not completely, reversible.
Minor points:
- The English grammar must be improved.
- Table 2 should be put into the supplements.
- “Line 81: The highest concentration of the solvent was <0.1%, and the solvent did not affect the measurements.”
How was this assessed?
- Line 184: ECGs?
- Figure 4: The amplitude and tau of dV/dtmax should be given and be statistically compared to the control. C and D should have the same scaling of the ordinate.
- Line 368: “at higher E. ciliata concentrations, harmonization of the impulse propagation directions (vectors) is observed (Figure 7)”.
Please explain in more detail.
Author Response
This is a well done study of the electrophysiological effect of Elsholtzia ciliata essential oil on the rabbit heart showing convincingly that the plant extract reduces the action potential (AP) amplitude, AP upstroke and AP propagation velocity leading to the conclusion that the extract exerts blocking effects on voltage gated Na channels like Ib type antiarrhythmic drugs. At higher concentrations, the extract slows down the AP propagation from the atria to the ventricles, which is probably due to antagonistic effects on voltage gated Ca2+ channels also.
First, we would like to thank Reviewer 1 for the evaluation of our manuscript as well as for the valuable remarks. Below are our answers to the specific remarks made by Reviewer 1. All changes of the previous version of the manuscript are highlighted in yellow.
However, I do have some problems with the interpretation of the data by the authors.
- The authors state that the herbs extracts may be harmless regarding side effects of antiarrhythmic drugs. It is well known that INa blockade with slowing of the AP propagation is potentially pro-arrhythmogenic. In this regard, I do not see any difference in the pro-arrhythmogenic effects of the herbs extract and antiarrhythmic chemicals lice lidocaine. The authors must discuss it.
Thank you for the comment. However, we do not state that E. ciliata is harmless. In the manuscript text we presented this as a presumption: "is supposed to be harmless" (page 2, line 43). We also discussed its pro-arrhythmic effect in the Discussion section (page 14).
Drugs blocking sodium current and slowing the AP propagation are attributed to the arrhythmic drugs, and they are potentially pro-arrhythmic, so possibly there is no big difference between herbs and chemicals like lidocaine.
We do not make any strict assumptions that there will be less harmful effects. However, based on the fact that preparation under investigation is of herbal origin, it is hoped that the drug may have a milder and less pro-arrhythmic effect.
In our study, we created conduction vector maps, which revealed that at high concentrations the vector length and direction of electrical signal propagation became more uniform (harmonized), compared to control and low concentrations of the E.ciliata and/or washout; and this might imply a less pro-arrhythmic effect.
- Any information about the plasma concentrations of the orally administered drug is needed to range the concentrations used in the study with regard to the therapeutic and side effects.
We appreciate this comment, and agree that pharmacokinetic features are important.
However, in this study we investigated the electrophysiological effects in order to reveal if the compound possesses 1st class antiarrhythmic features. Therefore, our aim was to investigate the pharmacodynamic but not pharmacokinetic features of E. ciliata.
Reviewer 1 wished to get information on the plasma concentrations of the orally administered compound. However, the most typical 1b class antiarrhythmic drug, the lidocaine, is almost never used for per-oral application.
At this stage we do not have information needed to range the used concentrations with regard to the therapeutic and side effects.
- Contrary to the statement of the authors, the measured electrophysiological effects are not, at least not completely, reversible.
We agree with the remark made by Reviewer 1. We used 20 min washout period and during this period recovery was not full. Therefore, statement “reversible” has been changed to “at least partially reversible” (p.1, line 30; p.7, line 183; p.10, line 241).
Minor points:
The English grammar must be improved.
Before resubmission, the revised manuscript has been edited by professional English editing service.
- Table 2 should be put into the supplements.
Thank you for suggestion, we put Table2 and Table3 into the supplements.
- “Line 81: The highest concentration of the solvent was <0.1%, and the solvent did not affect the measurements.”
How was this assessed?
As we stated in the Methods (page 3, line 85-86), stock solution with E. ciliata was prepared in anhydrous ethanol with volume ratio 1:1. The highest concentration of E. Ciliata was 0.3µL/mL, it means that the largest amount of ethanol was the same 0.3µL/mL, and by recalculation it is 0.024% concentration (density of ethanol 0.789).
The highest concentration of the solvent applied in our study was much lower than that suggested by Bebarova et al. (2010) to use in experiments on cardiomyocytes. Therefore, that reference [12] has been added in Section 2.2. “Chemicals” (page 3, line 88) and to the Reference list.
Adequately, the citation sequence in the manuscript text has been corrected.
- Line 184: ECGs?
We thank the Reviewer-1 for helping to notice the inaccuracy (page 7, line 195). This has now been corrected to “pseudo ECGs".
- Figure 4: The amplitude and tau of dV/dtmax should be given and be statistically compared to the control. C and D should have the same scaling of the ordinate.
Thank you for the comment. We included control data in the manuscript (page 11, lines 260).
The major information presented in Figure 4 is related to the changes in stimulation frequency, therefore the scale has been changed in order to reveal that information. We want to notice that it is important the frequency-dependent changes of dV/dtmax in percent, but not in the absolute values.
The control and E.ciliata calculated measurements have been added to the manuscript text (page 11, lines 260 and 265-266).
- Line 368: “at higher E. ciliata concentrations, harmonization of the impulse propagation directions (vectors) is observed (Figure 7)”.
Please explain in more detail.
We appreciate your comment and added information on impulse harmonization in the text of manuscript (p.13, lines 320-325):
“Under the control conditions and at low concentrations of E.ciliata extracts, as presented in Figure 7C, the direction and length of vectors vary, especially at the bottom of C‑vector maps. However at high concentrations, all vectors became of the same length and oriented in the same direction and this indicated harmonization in impulse propagation. Such uniform direction and the spread of electrical impulse propagation at high concentrations indicate that E. ciliata, alongside with the antiarrhythmic features, might have a less pro-arrhythmic effect.”
Submission Date
20 May 2020
02 Jun 2020 13:07:02 Date of this review
19 Jun 2020 16:50 Resubmission Date
Reviewer 2 Report
The paper by Mačianskienė et al. reports electrophysiological effects of Elsholtzia Ciliata essential oil found in Langendorff-perfused isolated rabbit heart. The results showed that E. ciliate essential oil affects cardiac electrophysiological properties in a way similar to the 1B class antiarrhythmic drugs, i.e. delayed activation and shortened action potential duration. The study presents solid data. My concerns are limited to the way of data presentation and interpretation and some technical issues.
Major comments.
General:
- The term “antiarrhythmic” used in the title and throughout the article does not correspond to the nature of the study since no arrhythmias were investigated. The manuscript reports the electrophysiological effects, which may be antiarrhythmic, but before direct testing, it remains unknown.
- In the present version, some parts of the material look excessive, and their necessity for the manuscript is not obvious. The authors should substantiate their presence or remove unnecessary parts. A) Was there any hypothesis to be tested by the application of different pacing modes? At first glance, all the findings can be drawn from just epicardial pacing, which permits analysis of individual action potentials as well as spatial parameters, such as conduction velocity. B) What was the purpose of both classical microelectrode and optical measurements of action potentials? If it is just a quality control approach, it would be logical to see it in the same subsection.
- There are parts of the text, which are misplaced. The Results section contains introductory and interpretative statements pertaining to the Methods or Discussion sections (lines 180-182, 186-194, 235-240, 287-291). Lines 264-267 largely duplicate the figure legend. Lines 110-111 are written in a way that the sentence looks like a description of the obtained data, not procedures used (as it is supposed to be).
- Check that all studied properties/parameters are referred to in the appropriate parts of the manuscript. In the present version, there is some inconsistency. A) Conduction velocity (and conduction vectors) is present in Fig. 7, not mentioned in the text of the Results, not described in the Methods. B) Time-to-Rmax is present in the Results, but not mentioned in the Methods.
Methods:
- Were the solvents added to control solutions or were their effects tested separately (line 82, Subsection 2.2. Chemicals)?
- To which ventricle epicardial stimulation was applied (lines 95-96)?
- Where electrical activity was recorded from? Was it left or right ventricle (lines 105, 119-120)? Was the interventricular groove captured in the mapped area?
- Lines 127-128. This description is not clear. A) QRS measurement. Both Q and S waves may be missing in recordings. It is supposed that the duration of the QRS complex is measured from its onset to its offset (whatever is the first wave and the last wave). However, it looks like (Fig.1) that it was Q-peak that was taken as the reference time-point. If it is the case, it is incorrect because such measurement misses the initial period of ventricular activation. B) QT measurement. The end-point of the QT interval was not defined clearly. The problem is that the T-wave in rabbits is often flattened, which terribly affects measurements. Was that manual or automatic detection? If automatic, what criteria were used? Actually, I would recommend (NOT necessary for this work) QTpeak measurement. It is more reliable, has a strong relation to minimal APD, and is nicely displayed in this article (Fig. 1).
Results:
- Figure 5 demonstrates an effect on activation time, not velocity. These two parameters are indeed related to each other in a general sense but not identical. The authors have direct data on conduction velocities presented in figure 7.
- Figure 7. What was the purpose of conduction velocity MAPPING vs calculation of the mean value? What was the purpose of the evaluation of conduction vectors?
Minor comments.
- Figure 2. Does the panel C display representative tracings or averaged? If representative, do they correspond to panel A?
- Figure 5 legend, line 276: 3 or 0.3?
- Check that all symbols are explained in the figure legends (*, #).
- Check Greek characters (or whatever it is supposed to be) (e.g. lines 113, 123).
Author Response
The paper by Mačianskienė et al. reports electrophysiological effects of Elsholtzia Ciliata essential oil found in Langendorff-perfused isolated rabbit heart. The results showed that E. ciliate essential oil affects cardiac electrophysiological properties in a way similar to the 1B class antiarrhythmic drugs, i.e. delayed activation and shortened action potential duration. The study presents solid data. My concerns are limited to the way of data presentation and interpretation and some technical issues.
First, we would like to thank the Reviewer-2 for acknowledging our study as solid data. We also thank for the explicit evaluation of our manuscript and for giving valuable comments. Please find point-by-point responses to each comment below. All changes made to the previous version of the manuscript are highlighted in yellow.
Major comments.
General:
- The term “antiarrhythmic” used in the title and throughout the article does not correspond to the nature of the study since no arrhythmias were investigated. The manuscript reports the electrophysiological effects, which may be antiarrhythmic, but before direct testing, it remains unknown.
We appreciate your comment and agree that no arrhythmias were investigated. Nevertheless, the blocking effect of the compound on the upstroke of action potential (AP) and the slowing down of the electrical signal conduction without doubt are properties characteristic of 1st class antiarrhythmic preparations. In this manuscript, we do not claim that E. ciliata can eliminate arrhythmias. There are many different mechanisms for the formation of arrhythmias and antiarrhythmic drugs have different effects.
In this study we investigate features that are generally rated as antiarrhythmic, such as blocking action on the AP upstroke and, as a consequence, a change of the electrical wave propagation and this is typical for 1st class antiarrhythmic drugs. All the substances which cause blocking action on the fast sodium current and on the AP upstroke are considered as having antiarrhythmic features.
In respect to here investigated features of the E ciliata which could be considered as antiarrhythmic, we prefer to keep the title without changes, because this reflects the aim and tasks of our study.
- In the present version, some parts of the material look excessive, and their necessity for the manuscript is not obvious. The authors should substantiate their presence or remove unnecessary parts.
- A) Was there any hypothesis to be tested by the application of different pacing modes? At first glance, all the findings can be drawn from just epicardial pacing, which permits analysis of individual action potentials as well as spatial parameters, such as conduction velocity.
Thank you for the comment, but we do not agree with such view, since different stimulation types allowed revealing abnormalities in the electrical signal conduction.
Besides, as was mentioned in the manuscript Results section (p.7-8, lines 196-206) and in Discussions (p.15, lines 390-393), this allow evaluation of electrical impulse propagation in different tissues and various directions. For example, only upon application of atrial stimulation a change in Ca2+-dependent atrio‑ventricular (AV) conduction was revealed, since the action potential (AP) in AV node is formed mainly by the L-type Ca2+-current. Epicardial stimulation allowed following electrical wave propagation in longitudinal direction alongside to the fiber orientation in the myocardium, and during endo- pacing (apex) we can follow the electrical signal propagation from the apex‑to‑base direction; under those pacing types only the sodium current may be responsible for possible changes in conduction. These peculiarities and the usefulness of applied three different stimulation types has been detailed in our earlier study by Macianskiene et al. (2015), which was cited in this manuscript.
- B) What was the purpose of both classical microelectrode and optical measurements of action potentials? If it is just a quality control approach, it would be logical to see it in the same subsection.
Reviewer-2 questioned the purpose of both classical microelectrode and optical measurements of action potentials, but their use was not just a quality approach.
As we demonstrated in our earlier study (Macianskiene et al., 2015) the upstroke of optical action potential does not correspond to the upstroke recorded with glass microelectrodes (compare Figures 5C vs 2A). In addition, marked differences in the upstroke shape have been revealed upon changing the stimulation type. The strength of optical mapping (OM) is the possibility to obtain APs in any place of field of view, and to follow the conduction processes of the electrical signal. Moreover, OM allow the construction of various maps across the entire heart surface (see Figures 7, 8, S1, S2, S3, S4 in this manuscript) or making movies (Martisiene et al. (Scientific Rep 2020, May)). Meanwhile classical glass-microelectrode recordings give a real shape of the AP upstroke, since a record is obtained from a single cell, and can be used for dV/dtmax and sodium current evaluations as a target of 1st class antiarrhythmic drugs. Nevertheless, recordings could be obtained just in the exact places where these microelectrodes are inserted, and the number of such microelectrodes is limited.
Considering the above information, we believe that conjunction of both techniques as well as combination of different stimulation types provides more complete information on electrical processes under investigation, which might be very important and interesting for the researchers working in this field.
- There are parts of the text, which are misplaced. The Results section contains introductory and interpretative statements pertaining to the Methods or Discussion sections (lines 180-182, 186-194, 235-240, 287-291). Lines 264-267 largely duplicate the figure legend. Lines 110-111 are written in a way that the sentence looks like a description of the obtained data, not procedures used (as it is supposed to be).
We agree with the notice that some parts in the manuscript are misplaced. According to Reviewer-2 suggestions, the corrected text is labeled in yellow.
- Check that all studied properties/parameters are referred to in the appropriate parts of the manuscript. In the present version, there is some inconsistency.
- A) Conduction velocity (and conduction vectors) is present in Fig. 7, not mentioned in the text of the Results, not described in the Methods.
Thank you for noticing missing information. As Reviewer-2 suggested, the conduction velocity and C-vectors have been described in Methods (page 5, lines 140-141) and added in Results (page 13, lines 314-325).
- B) Time-to-Rmax is present in the Results, but not mentioned in the Methods.
Thank you for noticing this inaccuracy; now this has been added in the Methods (page 5, lines 134): “Time-to-Rmax was detected as time to maximal amplitude to the R-peak from the stimulus.”
Methods:
- Were the solvents added to control solutions or were their effects tested separately (line 82, Subsection 2.2. Chemicals)?
Thank you for the comment. The solvent was not added to the control solutions because we used too small concentrations.
As we stated in the Methods (page 3, line 85-86), stock solution with E. ciliata was prepared in anhydrous ethanol with volume ratio 1:1. The highest concentration of E. Ciliata was 0.3µL/mL, it means that the largest amount of ethanol was the same 0.3µL/mL, and by recalculation it is 0.024% concentration (density of ethanol 0.789).
The highest concentration of the solvent applied in our study was much lower than that suggested by Bebarova et al. (2010) to use in experiments on cardiomyocytes. Therefore, that reference [12] has been added in Section 2.2. “Chemicals” (page 3, line 88) and to the Reference list.
Adequately, the citation sequence in the manuscript text has been corrected.
- To which ventricle epicardial stimulation was applied (lines 95-96)?
Thank you for noticing inaccuracy; this is now clarified as “to the left ventricle” (Corrected manuscript: page 4, line 102).
- Where electrical activity was recorded from? Was it left or right ventricle (lines 105, 119-120)? Was the interventricular groove captured in the mapped area?
Thank you for the comment. The electrical activity was obtained from a field of view of 20 x 20 mm, therefore captured LV, RV and the interventricular groove, and glass-microelectrodes were inserted in both LV and RV ventricles.
This information is added to the corrected manuscript: (page 4, line 113) “LV and RV”; (page 5, lines 126) “captured LV, RV and the interventricular groove”.
- Lines 127-128. This description is not clear.
- A) QRS measurement. Both Q and S waves may be missing in recordings. It is supposed that the duration of the QRS complex is measured from its onset to its offset (whatever is the first wave and the last wave). However, it looks like (Fig.1) that it was Q‑peak that was taken as the reference time-point. If it is the case, it is incorrect because such measurement misses the initial period of ventricular activation.
- B) QT measurement. The end-point of the QT interval was not defined clearly. The problem is that the T-wave in rabbits is often flattened, which terribly affects measurements. Was that manual or automatic detection? If automatic, what criteria were used? Actually, I would recommend (NOT necessary for this work) QTpeak measurement. It is more reliable, has a strong relation to minimal APD, and is nicely displayed in this article (Fig. 1).
Thank you for a valuable comment. ECG analysis was automatically produced using LabChart software for ECG analysis for rabbit heart.
T-wave was detected as time to half amplitude to the T-peak (page 5, lines 137). We also thank for the recommendation to use QT-peak measurement. We also have some observations in this field. Just one month ago we published the manuscript (Martisiene et al., 2020 in Scientific Reports), where we performed comparative evaluation of AP changes alongside with the ECG changes (in this case we evaluated ECG and RT-peak; Figures 4A, 4B in that manuscript). Data revealed that better correlation of the RT peak interval was obtained with the APD at 60% of repolarization. So, we can suggest that for evaluation of AP duration at 80% or 90% of repolarization might be more suitable the amplitude of 0.5 T-peak but not R-peak amplitude, and this is very important for the antiarrhythmic drugs.
Results:
- Figure 5 demonstrates an effect on activation time, not velocity. These two parameters are indeed related to each other in a general sense but not identical. The authors have direct data on conduction velocities presented in figure 7.
We agree with the Reviewer-2 comment, that activation time and conduction velocity is related to each other, but not identical, therefore we present both of them. Conduction velocity was calculated for ventricles. Activation time represents impulse propagation via both cardiac conduction system and AV node in the case of atrial pacing.
- Figure 7. What was the purpose of conduction velocity MAPPING vs calculation of the mean value? What was the purpose of the evaluation of conduction vectors?
We recorded conduction velocity and investigated the rate of propagation induced by E. ciliata.
Constructed CV maps demonstrate proportional changes under both the control conditions and after application of E. ciliata, and the maps show that there was no increase in the dispersion of this parameter.
C-vector maps show that direction of vectors after application of high concentrations of E. ciliata become more uniform, suggesting that at high concentrations of that herbal preparation propagation of electrical signal in the heart will be more harmonized. Accordingly, the E. ciliata possibly may be considered as having less pro-arrhythmic action.
This information has been added in the manuscript (page 13, lines 314-325).
Minor comments.
- Figure 2. Does the panel C display representative tracings or averaged? If representative, do they correspond to panel A?
Thank you for noticing missing information; this has now been added to the legend.
- Figure 5 legend, line 276: 3 or 0.3?
Thank you for noticing an inaccuracy, this has now been corrected as 0.3 (page 12, line 285).
- Check that all symbols are explained in the figure legends (*, #).
Thank you for noticing inaccuracies, all incorrect symbols have been corrected.
- Check Greek characters (or whatever it is supposed to be) (e.g. lines 113, 123).
Thank you for noticing inaccuracies, all incorrect symbols have been corrected.
Submission Date
20 May 2020
Date of this review
12 Jun 2020 10:04:41
19 Jun 2020 17:17 Resubmission Date
Reviewer 3 Report
The manuscript appears to be a continuation of authors' work on evaluation of biological effects of different preparations from Elsholtzia ciliate (e.g. recent publication in Molecules https://doi.org/10.3390/molecules25051153). Despite numerous biological activities known for constituents from plants of Elsholtzia gender (for review see: https://doi.org/10.1186/1752-153X-6-147), their antiarrhythmic properties were remained largely unexplored and authors’ finding and novel and interesting. Authors explore antiarrhythmic properties of Elsholtzia ciliate essential oil. The composition of the essential oil is similar to that reported by other groups (see Liu G, Wang H, Zhou BH, Song JC. GC-MS analysis of essential oil from Elsholtzia ciliata. Chin J Experim Trad Med Formulae. 2006; 12(11):18–21; Korolyuk EA, Koenig W, Tkachev AV. Composition of essential oils of Elsholtzia ciliata (Thunb.) Hyl. from the Novosibirsk region, Russia. Khimiya Rastitel'nogo Syr'ya. 2002; (1):31–36 and references cited there) with the main component dehydroelsholtsia ketone, which was first isolated from this source in 1997 and to which structure of 3-methyl-2-(3-methylbut-2-enoyl)furan was suggested (https://doi.org/10.1007/BF00629641). I think that the manuscript would have more value if the pure substance (dehydroelsholtsia ketone) was explored. The concentration of dehydroelsholtsia ketone in essential oils depends on many factors (see https://doi.org/10.29296/25877313-2020-04-01 and the references above) and therefore such interesting results on antiarrhythmic properties may not be very reproducible for similar preparations.
Nevertheless, biological investigations are performed at good level and well presented. Since no particular biomolecule is discussed in the manuscript, I feel that Biomolecules may not be an ideal journal for the publication of this otherwise interesting manuscript. I let Editor to make such decision on this overall reasonable submission, which requires some minor editing in language and formatting. Author may find the above references useful and can include them in the manuscript.
Author Response
The manuscript appears to be a continuation of authors' work on evaluation of biological effects of different preparations from Elsholtzia ciliate (e.g. recent publication in Molecules https://doi.org/10.3390/molecules25051153). Despite numerous biological activities known for constituents from plants of Elsholtzia gender (for review see: https://doi.org/10.1186/1752-153X-6-147), their antiarrhythmic properties were remained largely unexplored and authors’ finding and novel and interesting. Authors explore antiarrhythmic properties of Elsholtzia ciliate essential oil.
First, we would like to thank Reviewer-3 for the evaluation of our manuscript as well as for valuable suggestions. Below are answers to the specific remarks made by Reviewer‑3. All changes made to the previous version of the manuscript are highlighted in yellow.
The composition of the essential oil is similar to that reported by other groups (see Liu G, Wang H, Zhou BH, Song JC. GC-MS analysis of essential oil from Elsholtzia ciliata. Chin J Experim Trad Med Formulae. 2006; 12(11):18–21; Korolyuk EA, Koenig W, Tkachev AV. Composition of essential oils of Elsholtzia ciliata (Thunb.) Hyl. from the Novosibirsk region, Russia. Khimiya Rastitel'nogo Syr'ya. 2002; (1):31–36 and references cited there) with the main component dehydroelsholtsia ketone, which was first isolated from this source in 1997 and to which structure of 3-methyl-2-(3-methylbut-2-enoyl)furan was suggested (https://doi.org/10.1007/BF00629641).
We thank Reviewer-3 for finding interesting references. That information now has been added in the Introduction (p.2, lines 48-52):
“The composition of the essential oil is similar to that reported by other groups [6-8] with the main component of dehydroelsholtsia ketone, which was first isolated from this source in 1997 and to which a structure of 3-methyl-2-(3-methylbut-2-enoyl)furan had been suggested [9]. Chemical composition of E. ciliata herb could vary because of environmental factors and different growing places [10].”
I think that the manuscript would have more value if the pure substance (dehydroelsholtsia ketone) was explored. The concentration of dehydroelsholtsia ketone in essential oils depends on many factors (see https://doi.org/10.29296/25877313-2020-04-01 and the references above) and therefore such interesting results on antiarrhythmic properties may not be very reproducible for similar preparations.
Reviewer-3 is right. We agree that concentration of dehydroelsholtsia ketone in essential oils depends on many factors. Therefore, as Reviewer-3 proposed, investigation of the pure substance (dehydroelsholtsia ketone) might be helpful. However, in this study such investigation was not performed; nevertheless, this is a good idea which will be tested in the near future.
Nevertheless, biological investigations are performed at good level and well presented. Since no particular biomolecule is discussed in the manuscript, I feel that Biomolecules may not be an ideal journal for the publication of this otherwise interesting manuscript. I let Editor to make such decision on this overall reasonable submission, which requires some minor editing in language and formatting.
We appreciate the comment regarding the suitability of our manuscript to Biomolecules. However, this special issue was dedicated to “Perspectives of Essential Oils”. Besides, possible medical use of EOs has been highlighted as well. The E. ciliata as well as other Essential Oils substances might be important only upon revealing of its pharmacological features and its possible medical applicability is of particular importance.
Author may find the above references useful and can include them in the manuscript.
As Reviewer-3 suggested, new references [6-10] have been added to the Reference list. Adequately, the citation sequence in the manuscript text has been corrected.
Submission Date
20 May 2020
Date of this review
12 Jun 2020 15:50:10
19 Jun 2020 17:20 Resubmission Date